

**Evidence for mass independent fractionation of even**
**mercury isotopes in the troposphere**
Shuyuan Huang[1,2], Yunlong Huo[1], Heng Sun[1], Supeng Lv[1], Yuhan Zhao[1,2],
Kunning Lin[1], Yaojin Chen[2], Yuanbiao Zhang[1*]
*[1]Third Institute of Oceanography, Ministry of Natural Resources, Xiamen,*
*Fujian 361005, China*
*[2]State Key Laboratory of Marine Environmental Science, Xiamen*
*University, Xiamen, Fujian 361102, China*
*[*]Corresponding author: zhangyuanbiao@tio.org.cn*
**Abstract**: Mass independent fractionation (MIF) of even mercury (Hg)
isotopes has long been observed in atmospheric related samples and is
confirmed to be generated in the atmosphere, but its exact mechanism is
covered up by the Hg sources and atmospheric transformations and stays
unclear. Here, we present the first Hg isotope compositions of particulate
bound mercury (PBM) in the Northwest Pacific and observe highly
positive $\Delta^{200}$Hg values (up to 0.42‰). The MIF signatures are mainly
controlled by photoreduction, gaseous elemental mercury (GEM)
oxidation, and even-MIF dominated oxidation processes. Mercury in a
small part of samples influenced by anthropogenic emissions is recognized
by Hg concentrations and $\Delta^{199}$Hg signatures. The correlation between
$\Delta^{200}$Hg and light conditions confirms that even-MIF is linked to
photochemical reactions. The correlation between $\Delta^{200}$Hg and altitudes



suggests that a max even-MIF signatures existed in the troposphere. We
use $\Delta^{199}Hg/\Delta^{200}Hg$ ratios and ternary isotopic mixing model to estimate the
contributions of photoreduction, GEM oxidation and even-MIF dominated
oxidation. Our results demonstrate that atmospheric transformations are far
more important than Hg sources in shifting Hg isotope compositions of
PBM samples, especially in the marine boundary layer of the open ocean,
which is characterized by less anthropogenic influences and has
implications for our understanding of the mechanism of even-MIF and
subsequently Hg behaviors in the atmosphere.



## 1. Introduction

Mercury (Hg) mainly exists in three forms in the atmosphere: gaseous elemental mercury (GEM), gaseous oxidized mercury (GOM), and particulate bound mercury (PBM). GOM and PBM both exhibit a short residence time in the atmosphere and are readily deposited near Hg sources. In contrast, GEM is subject to long range transport and can be deposited significant distances from emission sources, including in relatively remote regions of the planet [1]. Atmospheric Hg continuously undergoes complicated physical and chemical transformations, including photoreactions, dark abiotic redox reaction, adsorption, and desorption, before being incorporated into the underlying surface [2]. Extensive efforts have been made to measure and model Hg in the atmosphere [3]. Recent advances in Hg stable isotopes have greatly improved our understanding of Hg cycling in the environment, including within the atmosphere [4-6].

Mercury has seven stable isotopes ($^{196}$Hg, $^{198}$Hg, $^{199}$Hg, $^{200}$Hg, $^{201}$Hg, $^{202}$Hg, and $^{204}$Hg). Mass dependent fractionation (MDF) of Hg isotopes is widely observed in natural environments, whereas significant mass independent fractionation of odd mass Hg isotopes (odd-MIF) is mainly induced by photochemical reactions [7]. Significant MIF of even mass Hg isotopes (even-MIF) has been measured in atmospheric samples or samples that relates to atmospheric Hg source [8-14]. However, the exact mechanism triggering even-MIF is unclear [15]. It is generally speculated that specific



oxidation reactions transforming Hg(0) to Hg(II) induced even-MIF. Fu et
al. attributed even-MIF to surface-mediated Hg photoreduction involving
halogens that can as of yet not explain [16]. Nonetheless, even-MIF
signatures have become a useful tracer to constrain atmospheric Hg
deposition pathways, reservoir size, and atmospheric conditions. Both
MDF and MIF provide useful information to trace Hg sources and identify
Hg transformation processes [13,16-18].
Isotopic compositions of PBM in urban and remote areas reflect
differing controls including anthropogenic emissions and atmospheric
transformations [19-23]. A recent study proposed that atmospheric
transformations could induce more positive odd-MIF values in PBM than
that from anthropogenic Hg sources [24]. Of marine boundary layer (MBL),
most PBM data have been collected from ground platforms [2] and show that
PBM plays an important role in the geochemical cycling of Hg. The MBL
is the largest transport layer and reaction vessel of atmospheric Hg due to
its relatively high humidity, sufficient sunlight, and abundant atmospheric
oxidants [25,26]. To date, several studies have been conducted in the MBL
over the open ocean and suggested that continental anthropogenic
emissions have little contribution to Hg in the open seas MBL and GEM
oxidation and photoreduction greatly shift isotopic compositions of Hg
species (i.e., GEM, PBM) in the MBL [17,27]. Hence, the Hg isotope
signatures in the MBL is of great significance to understand the





transformation of atmospheric Hg. However, the relative contributions of
GEM oxidation, photoreduction, and other atmospheric Hg transformation
processes is rarely quantified and reported.
This study investigated the distribution and source of PBM in the
MBL over the Northwest Pacific Ocean and the associated controlling
factors on even-MIF. PBM samples were collected during three cruises and
subsequently analyzed for selected Hg isotopes. The isotopic compositions
of the PBM were then combined with the Hybrid Single-Particle
Lagrangian Integrated Trajectory (HYSPLIT) model [28,29] to identify the
sources of PBM. Atmospheric transformations were identified by odd-MIF
and even-MIF. We proposed two ways to quantify contributions from
photoreduction, GEM oxidation, and even-MIF dominated photochemical
reactions in troposphere using Hg isotopic compositions.

**2.  Methods**
**2.1 Study sections.** Three Cruises were conducted in the Northwest Pacific
during the periods from August to November 2019 (denoted as Cruise A),
May to June 2018 (denoted as Cruise B), and August to September 2019
(denoted as Cruise C) (**Fig. 1**). Cruise A circumnavigated the southern
portion of the Northwest Pacific (7.00–24.74°N), whereas Cruise B
primarily navigated the western North Pacific around 30°N (21.00–
37.02°N). Cruise C was extended from the northwest marginal sea of the



North Pacific to the Bering Strait (34.52–73.34°N). Based on atmospheric
circulation and the results of 120 h back-trajectory analysis of air masses
from the HYSPLIT model, the northeast trade wind prevails at 0°N–30°N
and the westerly wind prevails at 30°N–60°N in the North Pacific. Polar
easterly winds prevail at high latitudes. The area of the South China Sea
and Western Pacific sampled during Cruise A is affected by the southwest
monsoon, while Northeast Asia, sampled during Cruise B, is controlled by
the northwest monsoon.

**2.2 Sample collection.** Two ARA N-FRM samplers (ARA Instruments,
USA) were deployed on the compass deck of the research vessel at a height
of approximately 15 m above mean sea level (AMSL) and at about 10 m
upstream of the exhaust outlet. To reduce the potential for contamination
from the ship's exhaust plume, sampling was stopped during station work
and when bad weather was encountered. Quartz fiber membranes (Grade
QMA, 47 mm, Whatman) were used for the collection of PBM as outlined
in a previous study [21]. The sampling time of each sample lasted for 48–72
h at a flow rate of about 20 L/min.

**2.3 Sample preconcentration.** PBM on the membrane was released via a
dual-stage tube furnace combustion protocol [21,30]. A $KMnO_4$ trapping
solution (0.1% $KMnO_4$ ($m/v$) + 10% $H_2SO_4$ ($v/v$)) was utilized to capture





the released Hg [21], after which it was preserved in the dark at 4°C until
analyzed for Hg concentration and relative isotopic abundances. Mercury
concentrations in the trapping solutions were measured by cold vapor
atomic fluorescence spectrometry (CVAFS, MERX, Brooks Rand
Instruments, USA) following US EPA Method 1631. PBM levels were so
low that Hg concentrations in the trapping solutions were lower than 1.0
ng/L. Hg standard solutions (NIST 3133) were therefore added to ensure
sufficient Hg mass (approximately 5 ng) for isotope analysis following our
previous study (**Note S1**) [31].

**2.4 Hg stable isotope analyses.** Mercury isotopic compositions of the
solutions were determined with a Nu Plasma Multi-Collector Inductively
Coupled Plasma Mass Spectrometer (MC-ICPMS) housed at the State Key
Laboratory of Marine Environmental Science at Xiamen University. A
modified cold-vapor generator and an Aridus III desolvating nebulizer
system (CETAC Technologies, USA) were used for Hg and thallium (Tl)
introduction, respectively, following previously published methods [32]. The
Hg isotopic compositions are reported in δ (‰) and Δ (‰) notation, which
represents the MDF and MIF of the Hg isotopes, respectively [33], such that:
$\delta^{xxx}Hg_{sample} = ((^{xxx}Hg_{sample}/^{198}Hg_{sample}) / (^{xxx}Hg_{NIST3133}/^{198}Hg_{NIST3133}) - 1) \times$

$1000$                                                                         (5)

$\Delta^{xxx}Hg = \delta^{xxx}Hg - \beta \times \delta^{202}Hg$                                      (6)



where xxx refers to the mass of each Hg isotope with atomic mass units
(amu) of 199, 200, 201, and 202. The fractionation factor $\beta$ is 0.2520,
0.5024, and 0.7520 for $^{199}$Hg, $^{200}$Hg, and $^{201}$Hg, respectively.
The repeated measurements of NIST 8610 gave long-term average
$\delta^{202}$Hg, $\Delta^{199}$Hg, and $\Delta^{200}$Hg values of $-0.52\pm0.12$‰ (2SD, n=5),
$-0.02\pm0.04$‰ (2SD, n=5), and $0.00\pm0.02$‰ (2SD, n=5), respectively.
These values are in accordance with previous studies [12,34,35]. Since the
samples were measured only once, the 2SD of the isotopic compositions
for each sample were selected as the 2SD of NIST 8610 (**Table S1**). The
isotopic compositions of PBM and the corresponding uncertainties (see
**Table S2**) were calculated by the method we previously developed [31].
Additional details pertaining to the approach are provided in the **Note S2**.

**2.5 Quality assurance and quality control.** Any unused pre-cleaned
quartz filters were placed in the closed sampling systems for 2 d to obtain
field blanks of PBM. In the process of sample preconcentration, unused
pre-cleaned quartz filters were periodically combusted with the samples to
obtain method blanks. For Hg isotopic analyses, the sample introduction
system was rinsed between samples with 3% $HNO_3$ solution until the signal
intensity dropped to background levels to avoid memory effects. Generally,
the blanks accounted for <3% in all the trapping solutions. The dual-stage
tube furnace combustion protocol was tested by adding NIST 3133 to



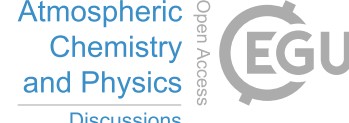

unused pre-cleaned quartz filters. The recovery was 101±8% (n=6, 1SD).

Standard solutions (NIST 8610) were used as a substitute for samples

and mixed with NIST 3133 in different proportions (33.3%, 50%, and
66.7%) to evaluate the isotopic compositions of PBM. The calculated
isotopic compositions of NIST 8610 were −0.57±0.10‰ (1SD, n=8),
−0.07±0.09‰ (1SD, n=8), and 0.01±0.04‰ (1SD, n=8) for $\delta^{202}$Hg, $\Delta^{199}$Hg,
and $\Delta^{200}$Hg, respectively (**Fig. S1**).

**2.6 Backward trajectories of air masses and identification of potential**
**source regions.** Air mass transport to the sampling area was assessed using
the NOAA Air Resources Laboratory GDAS 1° data archive and the
HYSPLIT model. Trajectory frequencies were analyzed to identify the
potential source as well as represent air mass transport. The trajectory
frequency started a trajectory from a single location and height every 6 h
and then summed the frequency that the trajectory passed over a grid cell.
This data was then normalized by the total number of trajectories (**Fig. S2**).
The parameter settings are shown in **Table S3**.

Locations of each sampling site are reported as the mean longitude

and latitude of the starting and ending points of sampling. The start time
used to run the model was chosen as the end date of sampling for each
sample (**Table S2**). The backward trajectories were calculated at a height
of 500 m AMSL and are representative of the approximate height of the



MBL where atmospheric pollutants are well mixed.

**2.7 Analyses of height of backward trajectories of air mass.** 315 h
backward trajectories of air mass arrived at 15 m height AMSL at each
sampling site were calculated at 1 h intervals using HYSPLIT model. Three
backward trajectories were output for each sampling site at starting point,
intermediate point, and end point. A total of 948 height values were
obtained for each sample. The frequency of trajectory points with height
higher than 500 m, 1500 m, and 3000 m AMSL ($f_{500}$, $f_{1500}$, and $f_{3000}$) were
calculated, respectively (**Table S4**). Furthermore, limit condition that
trajectory points fall in day light ($f_{500\_light}$, $f_{1500\_light}$, and $f_{3000\_light}$) was added
to represent air mass originated from upper atmosphere and suffered from
sunlight.

**2.8 Estimation of the max $\Delta^{200}$Hg values produced in the troposphere.**
For convenience, only three heights ($F_{position} < 1/18$, $1/18 < F_{position} < 1/6$,
$F_{position} > 1/6$) were chose to calculate the max $\Delta^{200}$Hg value based on the
following equations:
$$f_a \times \Delta_a^{200}Hg_i + f_b \times \Delta_b^{200}Hg_i + f_c \times \Delta_c^{200}Hg_i = \Delta_{sam\_iv}^{200}Hg \qquad (7)$$

$$f_a + f_b + f_c = 1 \qquad (8)$$

where $f_a$, $f_b$, and $f_c$ represent frequencies of trajectory points at a height of
$F_{position} < 1/18$, $1/18 < F_{position} < 1/6$, and $F_{position} > 1/6$, respectively.



$\Delta_a^{200}Hg_i$, $\Delta_b^{200}Hg_i$, $\Delta_c^{200}Hg_i$ represent $\Delta^{200}$Hg values at the three
heights and modeled by Monte Carlo simulation approach through the
pseudorandam number generation (i=1:10000). $\Delta_{sam\_iv}^{200}Hg$ represent
$\Delta^{200}$Hg values of PBM samples of type v.


## 3. Results

**3.1 PBM concentrations.** PBM concentrations exhibited mean values of

16.1±9.2 pg/m$^3$ (n=33, 2SD) and 16.0±10.5 pg/m$^3$ (n=7, 2SD) during
Cruises A and B, respectively (**Fig. 2a**). Concentrations were lower than
continental boundary layer PBM in most areas, especially in East Asia, but
higher than those in the MBL of the Pacific ocean (typically <10 pg/m$^3$) [2].
Long-term observations have shown that PBM concentrations are usually
lower than 30 pg/m$^3$ in the free troposphere [36], and thus, similar to that
found herein for the MBL.

Highly variable PBM concentrations (11.5–63.4 pg/m$^3$) were found in

samples from Cruise C; the mean concentration was 31.8±32.8 pg/m$^3$
(n=12, 2SD). The lowest concentration was observed at station 41, which
was in the Sea of Japan, while the highest concentration was observed at
station 47, which was in the Bering Sea. Compared to Cruises A and B,
Cruise C was closer to the coast. The elevated PBM concentrations in
Cruise C may thus be due to anthropogenic influences.





**3.2 PBM isotope compositions.** The isotope compositions of PBM
collected from the MBL exhibited negative $\delta^{202}Hg$ (−0.98±1.46‰, n=52,
2SD) and variable $\Delta^{199}Hg$ values (−0.08±0.56‰, n=52, 2SD). The isotopic
signatures overlapped with previously reported PBM isotope data. As
shown in **Fig. 3**, the PBM collected from urban regions exhibited large
variations in $\Delta^{199}Hg$ values (−0.02±0.61‰, n=205, 2SD) [19-23,37,38]. More
positive $\Delta^{199}Hg$ values (0.40±0.71‰, n=128, 2SD) have been observed for
PBM collected from areas of high altitude (from 50 m to 3816 m) [23,24]. The
PBM collected from polar regions displayed positive $\delta^{202}Hg$ and negative
$\Delta^{199}Hg$ values [11,13]. In contrast, the isotopic compositions of PBM collected
from specific emission sources, such as vehicles, industry, waste, flue gas,
and volcanos, have been characterized by negative $\delta^{202}Hg$ and near-zero
$\Delta^{199}Hg$ [5,39,40]. In addition, significant positive $\Delta^{200}Hg$ values (0.14±0.30‰,
n=52, 2SD) were observed in the current study (**Fig. 2d**).
Different isotopic signatures were found among different PBM
samples within the same cruise and between different cruises (**Fig. 2 & Fig.**
**S3**). Lower $\delta^{202}Hg$ values were observed near Micronesia and the Bering
Sea during Cruises A and C, while higher $\delta^{202}Hg$ values were found during
Cruise B (**Fig. 2b**). The spatial distribution of $\Delta^{199}Hg$ of PBM was more
complicated (**Fig. 2c**). For Cruises A and B, positive $\Delta^{199}Hg$ values were
mainly distributed in areas of open ocean in the Northwest Pacific, while
negative $\Delta^{199}Hg$ values were dispersed. Positive and negative $\Delta^{199}Hg$





values were observed during Cruise C in the marginal sea of Northeast Asia
and the Bering Sea, respectively.
**4. Discussion**
According to the $\Delta^{199}$Hg values, PBM samples could be divided into two
groups (**Fig. S4**): 1) samples characterized by positive $\Delta^{199}$Hg values,
which were related to the air mass from areas of open ocean (e.g., the
Northwest Pacific and the marginal sea in Northeast Asia), and 2) samples
possessing negative $\Delta^{199}$Hg values related to the air mass from the South
China Sea, Micronesia, Hawaii, Japan, Russia, and Bering Sea. Air mass
source analysis was predicted by HYSPLIT model (see Methods).
Long-range transport from anthropogenic emissions in mid-latitude
regions and GEM oxidation are two possible sources for the negative
$\Delta^{199}$Hg values of PBM [13]. The $\Delta^{199}$Hg values are plotted against Hg
concentrations in the PBM in **Fig. 4a**. Positive correlation are observed in
PBM samples with negative $\Delta^{199}$Hg values (**Fig. 4b**). The samples
collected during Cruises B and C possess a stronger correlation ($r^2$=0.61,
p<0.05) than the samples collected during Cruise A ($r^2$=0.04, p>0.1). The
higher Hg concentrations in PBM are most likely caused by the long-range
transport from anthropogenic emissions, which produced the slightly
negative $\Delta^{199}$Hg values. However, the significantly negative $\Delta^{199}$Hg values
associated with lower Hg concentrations in the PBM presumably result
from the *in situ* GEM oxidation. A few data points with slightly negative





$\Delta^{199}$Hg deviate positively from the regression lines in **Fig. 4b**. These
samples was likely influenced by the photoreduction of Hg(II), which
induced positive $\Delta^{199}$Hg values in residual Hg(II). Noted that the
background PBM pool is characterized by significant positive $\Delta^{199}$Hg
values, which is also associated with photoreduction [24]. Hence, samples
related to the open ocean with highly positive $\Delta^{199}$Hg values are mainly
controlled by photoreduction.

A large variation range of $\delta^{202}$Hg values are observed for PBM

samples with both positive and negative $\Delta^{199}$Hg values (**Fig. 4c**).
According to previous study, strong photoreduction of Hg(II) induced
positive shift of $\delta^{202}$Hg values in reactants (e.g., PBM) [27,41]. Although
inverse kinetic isotope effect was observed in Hg(0) oxidation by Cl and
Br atoms [42], the subsequent adsorption of Hg(II) on particulate surfaces
would lead to negatively shift of $\delta^{202}$Hg values in PBM. Furthermore, GEM
originated from dissolved gaseous mercury displayed more negative
$\delta^{202}$Hg (−2.98 to −0.99‰) [14] than that in the upper atmosphere (−0.02 to
1.64‰) [43,44], indicating that it is difficult to identify Hg source with $\delta^{202}$Hg
values. In addition to photochemical and redox reactions, gas-particle
partitioning, dissolution, and evaporation would also affect $\delta^{202}$Hg values
in PBM. Moreover, no correlation was found between $\delta^{202}$Hg values and
Hg concentrations ($p>0.1$, **Fig. 4d**). Therefore, the $\delta^{202}$Hg values cannot be
used as a diagnostic tool for atmospheric transformation processes.



However, the PBM samples with high Hg concentrations are characterized
by negative $\delta^{202}$Hg values, confirming the contribution from anthropogenic
sources. These samples are related to the air mass from Bering Sea,
coincide with enhanced levels of TGM observed in Bering Sea [45].
In addition to odd-MIF, the even-MIF is an important indicator for Hg
sources and transformation processes in the atmosphere [16]. As shown in
**Fig. 5**, the $\Delta^{200}$Hg values were considered to be contributed by
photochemical reactions in the upper troposphere. GEM oxidation could
also induce even-MIF at low altitudes, but limited to specific oxidants (e.g.,
Br, Cl). This process could be identified by backward air mass trajectory
as discussed below. The GEM oxidation processes without even-MIF, the
resulting PBM generally inherited even-MIF signatures of GEM and RGM
and was characterized by near-zero $\Delta^{200}$Hg values. Here, we define PBM
dominated by troposphere photochemical reactions are characterized by
odd-MIF and significant even-MIF ($\Delta^{200}$Hg≥0.15‰) and PBM dominated
by GEM oxidation without even-MIF at low altitudes are characterized by
negative odd-MIF and near-zero even-MIF at PBM. Therefore, except for
samples with high Hg concentrations (anthropogenic emissions), most
PBM samples were related with photoreduction (+odd-MIF), GEM
oxidation (–odd-MIF), and troposphere photochemical reactions (+even-
MIF), which could be identified by $\Delta^{199}$Hg and $\Delta^{200}$Hg values. Samples
related with GEM oxidation are overlapped with PBM collected from polar





region, which is also similar to GEM in terms of $\Delta^{199}$Hg and $\Delta^{200}$Hg values.
According to previous study, these samples are primarily sourced from
near-complete GEM oxidation [11,13]. As photoreduction induces no even-
MIF and troposphere photochemical reactions are considered to induce
even-MIF, samples with positive $\Delta^{199}$Hg and near-zero $\Delta^{200}$Hg values are
believed to experience photoreduction rather than photo-oxidation and vice
versa. Thus, the PBM samples could be classified into 6 types of different
sources and atmospheric transformations based on Hg concentrations and
isotope signatures (**Table 1**).
The ratio of $\Delta^{199}$Hg to $\Delta^{201}$Hg is commonly used to identify
fractionation processes. The $\Delta^{199}$Hg/$\Delta^{201}$Hg ratio was 0.75 and 0.81 for the
groups with positive and negative $\Delta^{199}$Hg values, respectively (**Fig. S5a**).
Both are lower than those observed for Hg(0) photo-oxidation (1.64–1.89)
[42] and Hg(II) photo-reduction (1.00–1.31) [41,46], but higher than that in TGM
from remote areas (0.73) [10,23,47] and coastal areas (0.55) [48,49]. The ratio
lower than 1.00 has been presented in oxic experiments during
photochemical reduction of Hg(II) [50]. These ratios indicated that the odd-
MIF of PBM isotopes was induced by multiple processes rather than a
single oxidation or reduction process. For each cruise, the ratio increased
in the order Cruise A > Cruise B > Cruise C (**Fig. S5b**). With regards to the
variations in latitude among the three cruises (**Table S2**), the ratios have a
statistical correlation with mean latitude of each cruise ($R^2$=0.99, $p$<0.01).



However, no correlations between the ratios and the corresponding mean
$\Delta^{199}$Hg value was found, which has been observed in PBM samples with
highly positive $\Delta^{199}$Hg from megacities [51]. Also, no correlations between
odd-MIF and latitude for PBM were found ($p>0.05$). Similarly, no
correlations between odd-MIF and radiation duration for PBM were
observed ($p>0.05$, the radiation duration was represented by astronomical
twilight which could be obtained from websites www.wunderground.com),
which is inconsistent with a previous study [20] and implies potential impact
of oxidations on the odd-MIF. It has been previously suggested that
troposphere photochemical reactions that caused even-MIF was possibly
accompanied by odd-MIF, although the magnitude is smaller than that
induced by photoreduction [16]. Another reason maybe is that this study used
ship-based sampling rather than fixed-point sampling, the baseline
(background) PBM isotope signatures varied with sampling sites.
Additionally, latitude was closely related with air temperature and light
intensity, which may affect the MIF of PBM, but air temperature and light
intensity were absent in this study. Nonetheless, the ratio for all PBM
samples was 1.02 in this study, suggested odd-MIF was mainly caused by
photochemical reactions and $\Delta^{199}$Hg was a comprehensive result of
different transformations.
The mechanism of even-MIF has been discussed for nearly a decade.
Most scientists support the view that even-MIF is produced in the upper





atmosphere and could be used to estimate the contribution of Hg from the
upper troposphere [11,12,15,24]. Recent studies, however, propose a different
mechanism in which small but significant $\Delta^{200}$Hg anomalies can originate
from *in situ* photooxidation of Hg(0) by UV light and oxidants at low
altitudes [16,24,42]. Thus, we traced the air mass height of PBM samples during
the past 13 days in comparison to the moment of sample collection (**Fig.**
**S6**). This timeframe was selected because the global mean lifetime of Hg(II)
before photoreduction in the troposphere is approximately 13 days [52]. The
back trajectory model showed that air masses mainly (>90%) originated
from below 1.5 km AMSL before arriving at the sampling sites in 21 out
of 52 of the PBM samples, while air masses were completely derived from
below 500 m AMSL in only 3 PBM samples (see **Methods; Table S4**).
Highly positive $\Delta^{200}$Hg values (≥0.15‰) were observed in 24 PBM
samples. Discrete data points exhibited $\Delta^{200}$Hg values that ranged from
−0.11‰ to 0.14‰ (n=28). Here, we analyzed the relationships between
$\Delta^{200}$Hg values and frequencies of trajectory points when heights were
higher than 500 m, 1500 m, and 3000 m AMSL ($f_{500}$, $f_{1500}$, and $f_{3000}$) (**Fig.**
**6a, b, and c**). Air masses mainly originated from low altitudes during
Cruise A, because the mean frequencies were higher in Cruises B and C
(48.08% for $f_{500}$ and 27.17% for $f_{1500}$) than in Cruise A (25.33% for $f_{500}$ and
11.41% for $f_{1500}$). Consequently, the latter mechanism in which even-MIF
produced at low altitudes was dominant in shaping $\Delta^{200}$Hg values of the





PBM samples is supported by three lines of evidence: (1) highly positive
$\Delta^{200}Hg$ values were positively correlated with multiple frequencies (i.e.,
$f_{500}, f_{1500}, f_{3000}$) during Cruise A, and the correlations were more significant
at a lower altitude; (2) the positive relationships between highly positive
$\Delta^{200}Hg$ values and frequencies $f_{500}$ and $f_{1500}$ were stronger in Cruise A than
in Cruises B and C; and (3) significant negative relationships were
observed between highly positive $\Delta^{200}Hg$ values and $f_{3000}$ in Cruises B and
C with more air masses from high altitudes. Moreover, the highly positive
$\Delta^{200}Hg$ values in 6 PBM samples displayed $f_{1500} = 0$, indicating that these
samples were associated with an air mass from low altitudes (<1.5 km) as
the even-MIF in these samples was mainly produced at low altitudes. The
mechanism could be due to vertical mixing of air masses from high and
low altitudes, or photo-oxidation of Hg(0) at low altitudes. We also
analyzed the frequencies as a function of height and light conditions; that
is, when trajectory points occurred during day light ($f_{500\_light}$, $f_{1500\_light}$, and
$f_{3000\_light}$) (**Fig. 6d, e, and f**). The results showed that light played an
important role in even-MIF, as $\Delta^{200}Hg$ and $f_{500\_light}$ ($f_{1500\_light}$, $f_{3000\_light}$)
exhibited an analogous correlation to $\Delta^{200}Hg$ and $f_{500}$ ($f_{1500}, f_{3000}$) ($p>0.05$,
paired $t$ test).
There would be a height that produce the max $\Delta^{200}Hg$ value in terms
of the negative relationship between $f_{3000}$ and $\Delta^{200}Hg$, although the trend
between $f_{3000}$ and $\Delta^{200}Hg$ is completely opposite in Cruise A and in Cruise





B and C, that is, the trend is opposite at low and high latitudes (**Fig. 6c**). In
view of the fact that the tropospheric height decreases with the increase of
latitude, the position of the same altitude relative to the tropopause ($F_{\text{position}}$)
is different at high and low latitudes. For examples, we set the tropopause
height at high and low latitudes to 9 km and 18 km, the position of 3000 m
is 1/3 and 1/6 of the tropopause, respectively. Hence, we could obtain
$F_{\text{position}}$ = height/9000 at high latitudes and $F_{\text{position}}$ = height/18000 at low
latitudes, respectively. Fit $F_{\text{position}}$ to the slope of $\Delta^{200}$Hg and $f_{500}$ ($f_{1500}, f_{3000}$)
linearly (**Fig. S7**), the slope showed a negative linear correlation of the
$F_{position}$, indicating that, as the height increases, the $\Delta^{200}$Hg values first
increases and then decreases. Hence, there should be a maximum $\Delta^{200}$Hg
values at a certain height. The fitting curve ($r^2$=0.95, $p$<0.01) intersects the
x axis at 0.18, indicating that the max $\Delta^{200}$Hg value would be produced at
$F_{\text{position}}$ = 0.18, that is, the max $\Delta^{200}$Hg value would be produced at the
height of 3240 m and 1620 m at low and high latitudes, respectively. Thus,
the results confirmed the opposite trend between $f_{3000}$ and $\Delta^{200}$Hg at low
and high latitudes. The max $\Delta^{200}$Hg value was estimated by a Monte Carlo
simulation approach and isotopic mixing model using data of type v which
is dominated by troposphere photochemical reactions (see **Methods**). The
estimated value of the max $\Delta^{200}$Hg of 1.10±0.58 for PBM is overall similar
to the maximum value (1.24 ± 0.08) currently observed in atmospheric
environment [8].


The precipitation samples are characterized by more positive $\Delta^{200}Hg$
values than PBM and GOM samples [16,17,27,53], suggesting that even-MIF
produced in the MBL over the open ocean tend to be accumulated in
aqueous phase and brought into seawater. The open ocean seawater shows
positive $\Delta^{200}Hg$ values [54,55]. Latitudinal variations of $\Delta^{200}Hg$ were observed
in seawater and precipitation, although negative correlation was found in
seawater $\Delta^{200}Hg$ and latitudes and positive correlation was found in
precipitation $\Delta^{200}Hg$ and latitudes, respectively [55,56]. As suggested,
seawater $\Delta^{200}Hg$ are controlled by both Hg(0) and Hg(II) and larger ocean
Hg(0) uptake at high latitudes results in low seawater $\Delta^{200}Hg$ [55]. While for
precipitation $\Delta^{200}Hg$, max $\Delta^{200}Hg$ values in the troposphere are estimated
to be generated in lower altitudes at high latitudes, $\Delta^{200}Hg$ accumulated in
precipitations would be less affected before reaching the ground (being
collected).
As discussed above, photoreduction, troposphere photochemical
reactions, and GEM oxidation are the three major processes that trigger
odd-MIF and even-MIF (**Table 1**). Noted that adsorption and desorption
between PBM and GOM induced no odd-MIF and the sources of Hg in the
atmosphere are accompanied with near-zero odd-MIF [5,39,40]. Moreover, the
$\Delta^{200}Hg$ values for GEM, GOM, and PBM are near-zero in source materials
and therefore emissions [57]. Therefore, the $\Delta^{200}Hg$ and $\Delta^{199}Hg$ values could
be used to estimate the contributions of photoreduction, photo-oxidation,



and GEM oxidation. Here, we proposed two ways to quantify contributions
from atmospheric transformations: (a) specific diagnostic ratios of
$\Delta^{199}Hg/\Delta^{200}Hg$; (b) ternary isotopic mixing model.
(a) Even-odd patterns of Hg isotope fractionation factors have been
proposed to be a better indicator of MIF mechanisms than $\Delta^{199}Hg/\Delta^{201}Hg$
slopes. According to previous study, a consistent pattern between $\Delta^{199}Hg$
and $\Delta^{200}Hg$ has been observed with large variation range (1.1-3.3) on GEM
and oxidized Hg phases (i.e., reactive Hg and precipitation Hg) [16,48,58]. As
listed in Table 1, samples in type i could be used to represent near-complete
GEM oxidation and inherit the isotope composition of GEM. We then built
$\Delta^{199}Hg/\Delta^{200}Hg$ slopes between type i and type ii, iii, and v, respectively.
Based on Williamson-York bivariate linear regression method [59], the
observed fitted curve shaped a slope of 1.81±0.27 ($r^2$=0.57, $p$<0.01),
4.90±0.82 ($r^2$=0.31, $p$<0.05), and 0.03±0.16 ($r^2$=0.00, $p$=0.94), respectively.
We suggest that the slope was primarily determined by the isotope
signatures of oxidized Hg phases, because GEM varies in a narrow range
of $\Delta^{199}Hg$ and $\Delta^{200}Hg$ with -0.20±0.17‰ and -0.06±0.12‰ (2SD, n=208),
respectively [9,10,13,16,44,48,58,60-63]. Thus, we hypothesized that the ratios of
$\Delta^{199}Hg/\Delta^{200}Hg$ could be used as a diagnostic for proportions of
photoreduction and troposphere photochemical reactions despite of GEM
oxidation. To gain a more representative result, we integrate our data with
a subset of the published PBM data (**Fig. 7**). The resulting ratios of





$\Delta^{199}Hg/\Delta^{200}Hg$ were $17.71\pm1.42$ ($r^2=0.27$, $p<0.01$) and $0.61\pm0.06$ ($r^2=0.43$,
$p<0.01$) for photoreduction-dominated and troposphere-oxidation-
dominated PBM samples, respectively. The proportions of photoreduction
and photo-oxidation could be calculated as the equation below:
$$f_{red} = \frac{\arctan(k)-\arctan 0.61}{\arctan 17.71 - \arctan 0.61} \times 100\% = \frac{\arctan(k)-0.548}{0.966} \times 100\% \qquad (1)$$
where $f_{red}$ represents proportions of photoreduction and k represent ratios
of $\Delta^{199}Hg/\Delta^{200}Hg$. Following equation (1), we suggested that $66\pm16\%$ of
Hg was photoreduced in precipitation and $75\pm16\%$ Hg was photoreduced
in oxidized Hg phases at high altitudes (**Table S6**). In this study, the $f_{red}$
was calculated for sample types ii, iii, and v and listed in **Table 1**. It should
be noted again that the $f_{red}$ was calculated without considering the influence
of GEM oxidation. If GEM oxidation effects greatly, such as the observed
$\Delta^{199}Hg$ values are near-zero or negative, the ratios would be useless. For
example, the calculated $f_{red}$ for sample types v was -0.58 and meaningless.

(b) As shown in **Fig. 7**, the three end-member could be used to

calculate contributions from photoreduction, troposphere photochemical
reactions, and GEM oxidation through ternary isotopic mixing model.
$$X \cdot \Delta^{200}Hg_{red} + Y \cdot \Delta^{200}Hg_{oxi} + Z \cdot \Delta^{200}Hg_{GEMo} = \Delta^{200}Hg_{sam} \qquad (2)$$
$$X \cdot \Delta^{199}Hg_{red} + Y \cdot \Delta^{199}Hg_{oxi} + Z \cdot \Delta^{199}Hg_{GEMo} = \Delta^{199}Hg_{sam} \qquad (3)$$
$$X + Y + Z = 1 \qquad (4)$$
where X, Y, and Z represent the contribution of the three atmospheric
transformations of the photoreduction, troposphere photochemical





reactions, and GEM oxidation, respectively. The Hg isotope signatures of
the three atmospheric transformations are the mean values of the three end-
member as mentioned above (**Table S5**). The results of the mixing model
calculation showed that the contributions of photoreduction, troposphere
photochemical reactions, and GEM oxidation varied within -12% to 77%,
-12% to 35%, and 2% to 97%, respectively (**Table 1**). The results showed
similar contribution from photoreduction with estimations from ratios of
$\Delta^{199}Hg/\Delta^{200}Hg$. The estimated highly contributions (53±22%, 1SD) of
GEM oxidation is consistent with previous suggestion that GEM oxidation
played a major role (~47±22%) in the formation of Hg(II) in PBM [17,51]. The
weak    contributions    (10±12%,    1SD)    of    troposphere    photochemical
reactions indicate strong dilutions of near-zero $\Delta^{200}Hg$ from Hg sources
and other atmospheric transformations.

**5. Conclusion**

The isotopic compositions of PBM in the MBL of the Northwest

Pacific suggest that the even-MIF and odd-MIF signatures are useful
tracers for identifying atmospheric transformations. The $\delta^{202}Hg$ signature
was significantly shifted by Hg sources and *in situ* transformations of
atmospheric Hg. Strong correlations between the even-MIF of PBM and
height of the air mass provide additional support for the occurrence of
even-MIF in the troposphere, which may link to photochemical reactions



in the atmosphere. The mechanism driving the highly positive $\Delta^{200}$Hg
signatures of PBM in this study is probably the oxidation of Hg(0) at low
altitudes. The MBL over the open ocean may promote the intrusion of
even-MIF signatures and its subsequent recording by seawater and marine
sediments [55,64,65]. It is highly uncertain how strong even-MIF signatures
could be generated in the troposphere, the estimated max $\Delta^{200}$Hg value
provides data support for future research. The quantification of
contributions of atmospheric transformations by the proposed methods
have implications for the study of Hg behaviors in the atmosphere.

**Description of statistical analysis.** All the statistical analyses were
performed using Origin 2019b and Excel 2019. The Paired *t*-test, Pearson's
R-Square and *P*-value are calculated by algorithms of the software.

**Data availability.** The authors declare that the main data supporting the
findings of this study are available within the paper and its supplementary
information files. Extra data are available from the corresponding author
upon request.

**Author contributions**
S.H., and Y.Z.* conceived and designed this project, S.H., Y.H., and H.S.
conducted the field sampling, S.H., S.L., Y.Z., and Y.C. carried out all the



measurements, and S.H., K.L., and Y.Z.[*] wrote the draft paper.

**Competing interests**

The authors declare no competing interests.

**Additional information**

Correspondence and requests for materials should be addressed to Y.Z.

**Acknowledgments**

This research was financed by the National Science Foundation for
Young Scientists of China (22006168), the National Key Research and
Development Program of China (2019YFA0607003), the Chinese Projects
for Investigations and Assessments of the Arctic and Antarctic
(CHINARE2017-2021), and the Natural Science Foundation of Fujian
Province, China (2020J05074). The authors acknowledge the support of
the Fujian Science and Technology Innovation Leader Project 2016. We
thank LetPub (www.letpub.com) for linguistic assistance and pre-
submission expert review.

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

spatiotemporal variations of atmospheric speciated mercury: a review. *Atmos. Chem. Phys.*





| 566 | | **16**, 12897-12924, doi:10.5194/acp-16-12897-2016 (2016). |
|---|---|---|
| 567 | 3 | Gustin, M. S., Amos, H. M., Huang, J., Miller, M. B. & Heidecorn, K. Measuring and modeling |
| 568 | | mercury in the atmosphere: a critical review. *Atmos. Chem. Phys.* **15**, 5697-5713, |
| 569 | | doi:10.5194/acp-15-5697-2015 (2015). |
| 570 | 4 | Sun, R. *et al.* Modelling the mercury stable isotope distribution of Earth surface reservoirs: |
| 571 | | Implications for global Hg cycling. *Geochim. Cosmochim. Acta* **246**, 156-173 (2018). |
| 572 | 5 | Das, R. *et al.* Mercury isotopes of atmospheric particle bound mercury for source |
| 573 | | apportionment study in urban Kolkata, India. *Elementa-Sci. Anthrop.* **4**, 12 (2016). |
| 574 | 6 | Douglas, T. & Blum, J. Mercury Isotopes Reveal Atmospheric Gaseous Mercury Deposition |
| 575 | | Directly to the Arctic Coastal Snowpack. *Environ. Sci. Technol. Let.* **6**, 235-242 (2019). |
| 576 | 7 | Blum, J., Sherman, L. & Johnson, M. Mercury Isotopes in Earth and Environmental Sciences. |
| 577 | | *Annu. Rev. Earth Planet. Sci.* **42**, 249-269 (2014). |
| 578 | 8 | Chen, J., Hintelmann, H., Feng, X. & Dimock, B. Unusual fractionation of both odd and |
| 579 | | even mercury isotopes in precipitation from Peterborough, ON, Canada. *Geochim.* |
| 580 | | *Cosmochim. Acta* **90**, 33-46 (2012). |
| 581 | 9 | Rolison, J., Landing, W., Luke, W., Cohen, M. & Salters, V. Isotopic composition of species- |
| 582 | | specific atmospheric Hg in a coastal environment. *Chem. Geol.* **336**, 13 (2013). |
| 583 | 10 | Demers, J., Blum, J. & Zak, D. Mercury isotopes in a forested ecosystem: Implications for |
| 584 | | air-surface exchange dynamics and the global mercury cycle. *Global Biogeochem. Cycles* |
| 585 | | **27**, 222-238 (2013). |
| 586 | 11 | Li, C. *et al.* Seasonal Variation of Mercury and Its Isotopes in Atmospheric Particles at the |
| 587 | | Coastal Zhongshan Station, Eastern Antarctica. *Environ Sci Technol* **54**, 11344-11355, |
| 588 | | doi:10.1021/acs.est.0c04462 (2020). |
| 589 | 12 | Jiskra, M., Sonke, J., Agnan, Y., Helmig, D. & Obrist, D. Insights from mercury stable |
| 590 | | isotopes on terrestrial–atmosphere exchange of Hg(0) in the Arctic tundra. |
| 591 | | *Biogeosciences* **16**, 4051-4064 (2019). |
| 592 | 13 | Zheng, W. *et al.* Mercury stable isotopes reveal the sources and transformations of |
| 593 | | atmospheric Hg in the high Arctic. *Appl. Geochem.* **131**, 105002, |
| 594 | | doi:10.1016/j.apgeochem.2021.105002 (2021). |
| 595 | 14 | Huang, S., Lin, K., Yuan, D., Gao, Y. & Sun, L. Mercury isotope fractionation during transfer |
| 596 | | from post-desulfurized seawater to air. *Mar. Pollut. Bull.* **113**, 81-86 (2016). |
| 597 | 15 | Cai, H. & Chen, J. Mass-independent fractionation of even mercury isotopes. *Sci. Bull.* **61**, |
| 598 | | 116-124 (2016). |
| 599 | 16 | Fu, X. *et al.* Mass-Independent Fractionation of Even and Odd Mercury Isotopes during |
| 600 | | Atmospheric Mercury Redox Reactions. *Environ Sci Technol* **55**, 10164-10174, |
| 601 | | doi:10.1021/acs.est.1c02568 (2021). |
| 602 | 17 | Yu, B. *et al.* New evidence for atmospheric mercury transformations in the marine |
| 603 | | boundary layer from stable mercury isotopes. *Atmos. Chem. Phys.* **20**, 9713-9723, |
| 604 | | doi:10.5194/acp-20-9713-2020 (2020). |
| 605 | 18 | Meng, M. *et al.* Mercury Inputs Into Eastern China Seas Revealed by Mercury Isotope |
| 606 | | Variations in Sediment Cores. *Journal of Geophysical Research: Oceans* **126**, |
| 607 | | doi:10.1029/2020jc016891 (2021). |
| 608 | 19 | Qiu, Y. *et al.* Identification of potential sources of elevated PM2.5-Hg using mercury |
| 609 | | isotopes during haze events. *Atmos. Environ.* **247**, 118203, |





| 610 | | doi:10.1016/j.atmosenv.2021.118203 (2021). |
| 611 | 20 | Huang, Q. *et al.* Diel variation of mercury stable isotope ratios record photoreduction of |
| 612 | | PM2.5-bound mercury. *Atmos. Chem. Phys.*, 315-325 (2019). |
| 613 | 21 | Huang, S. *et al.* Natural stable isotopic compositions of mercury in aerosols and wet |
| 614 | | precipitations around a coal-fired power plant in Xiamen, southeast China. *Atmos. Environ.* |
| 615 | | **173**, 72-80 (2018). |
| 616 | 22 | Xu, H. *et al.* Mercury stable isotope compositions of Chinese urban fine particulates in |
| 617 | | winter haze days: Implications for Hg sources and transformations. *Chem. Geol.* **504**, 267- |
| 618 | | 275 (2018). |
| 619 | 23 | Yu, B. *et al.* Isotopic Composition of Atmospheric Mercury in China: New Evidence for |
| 620 | | Sources and Transformation Processes in Air and in Vegetation. *Environ. Sci. Technol.* **50**, |
| 621 | | 9262-9269 (2016). |
| 622 | 24 | Fu, X. *et al.* Domestic and Transboundary Sources of Atmospheric Particulate Bound |
| 623 | | Mercury in Remote Areas of China: Evidence from Mercury Isotopes. *Environ. Sci. Technol.* |
| 624 | | **53**, 1947-1957 (2019). |
| 625 | 25 | Hedgecock, I. & Pirrone, N. Chasing quicksilver: Modeling the atmospheric lifetime of Hg0 |
| 626 | | (g) in the marine boundary layer at various latitudes. *Environ. Sci. Technol.* **38**, 69–76 |
| 627 | | (2004). |
| 628 | 26 | Hedgecock, I. & Pirrone, N. Mercury and photochemistry in the marine boundary layer- |
| 629 | | modelling studies suggest the in situ production of reactive gas phase mercury. *Atmos.* |
| 630 | | *Environ.* **35**, 3055–3062 (2001). |
| 631 | 27 | Qiu, Y. *et al.* Stable mercury isotopes revealing photochemical processes in the marine |
| 632 | | boundary layer. *J. Geophys. Res.: Atmos.*, doi:10.1029/2021jd034630 (2021). |
| 633 | 28 | Stein, A. F. *et al.* NOAA's HYSPLIT Atmospheric Transport and Dispersion Modeling System. |
| 634 | | *Bull. Am. Meteorol. Soc.* **96**, 2059-2077, doi:10.1175/bams-d-14-00110.1 (2015). |
| 635 | 29 | Rolph, G., Stein, A. & Stunder, B. Real-time Environmental Applications and Display |
| 636 | | sYstem: READY. *Environ. Modell. Softw.* **95**, 210-228, doi:10.1016/j.envsoft.2017.06.025 |
| 637 | | (2017). |
| 638 | 30 | Huang, Q. *et al.* An improved dual-stage protocol to preconcentrate mercury from |
| 639 | | airborne particles for precise isotopic measurement. *J. Anal. At. Spectrom.* **30**, 966 (2015). |
| 640 | 31 | Huang, S. *et al.* Application of an isotope binary mixing model for determination of precise |
| 641 | | mercury isotopic composition in samples with low mercury concentration. *Anal. Chem.* **91**, |
| 642 | | 7063-7069 (2019). |
| 643 | 32 | Lin, H. *et al.* Isotopic composition analysis of dissolved mercury in seawater with purge |
| 644 | | and trap preconcentration and a modified Hg introduction device for MC-ICPMS. *J. Anal.* |
| 645 | | *At. Spectrom.* **30**, 353-359 (2015). |
| 646 | 33 | Blum, J. & Bergquist, B. Reporting of variations in the natural isotopic composition of |
| 647 | | mercury. *Anal. Bioanal. Chem.* **388**, 359 (2007). |
| 648 | 34 | Janssen, S. E. *et al.* Examining historical mercury sources in the Saint Louis River estuary: |
| 649 | | How legacy contamination influences biological mercury levels in Great Lakes coastal |
| 650 | | regions. *Sci. Total Environ.* **779**, 146284, doi:10.1016/j.scitotenv.2021.146284 (2021). |
| 651 | 35 | Enrico, M., Balcom, P., Johnston, D. T., Foriel, J. & Sunderland, E. M. Simultaneous |
| 652 | | combustion preparation for mercury isotope analysis and detection of total mercury using |
| 653 | | a direct mercury analyzer. *Anal. Chim. Acta* **1154**, 338327, doi:10.1016/j.aca.2021.338327 |





(2021).

Timonen, H., Ambrose, J. L. & Jaffe, D. A. Oxidation of elemental Hg in anthropogenic and
marine airmasses. *Atmos. Chem. Phys.* **13**, 2827-2836, doi:10.5194/acp-13-2827-2013
(2013).

Huang, Q., Reinfelder, J., Fu, P. & Huang, W. Variation in the mercury concentration and
stable isotope composition of atmospheric total suspended particles in Beijing, China. *J.*
*Hazard. Mater.* **383**, 121131 (2019).

Huang, Q. *et al.* Isotopic composition for source identification of mercury in atmospheric
fine particles. *Atmos. Chem. Phys.* **16**, 14 (2016).

Li, X. *et al.* Isotope signatures of atmospheric mercury emitted from residential coal
combustion. *Atmos. Environ.* **246**, 118175, doi:10.1016/j.atmosenv.2020.118175 (2021).

Zambardi, T., Sonke, J., Toutain, J., Sortino, F. & Shinohara, H. Mercury emissions and
stable isotopic compositions at Vulcano Island (Italy). *Earth and Planetary Science Letters*
**277**, 236-243 (2009).

Bergquist, B. & Blum, J. Mass-Dependent and -Independent Fractionation of Hg Isotopes
by Photoreduction in Aquatic Systems. *Science* **318**, 417-420 (2007).

Sun, G. *et al.* Mass-Dependent and -Independent Fractionation of Mercury Isotope during
Gas-Phase Oxidation of Elemental Mercury Vapor by Atomic Cl and Br. *Environ. Sci.*
*Technol.* **50**, 9232-9241 (2016).

Fu, X. *et al.* Isotopic compositions of atmospheric total gaseous mercury in ten Chinese
cities and implications for land surface emissions. *Atmos. Chem. Phys.*, Preprint,
doi:10.5194/acp-2020-981 (2021).

Fu, X., Maruszak, N., Wang, X., Gheusi, F. & Sonke, J. Isotopic Composition of Gaseous
Elemental Mercury in the Free Troposphere of the Pic du Midi Observatory, France.
*Environ. Sci. Technol.* **50**, 5641-5650 (2016).

Kang, H. & Xie, Z. Atmospheric mercury over the marine boundary layer observed during
the third China Arctic Research Expedition. *Journal of Environmental Sciences* **23**, 1424-
1430, doi:10.1016/s1001-0742(10)60602-x (2011).

Zheng, W. & Hintelmann, H. Mercury isotope fractionation during photoreduction in
natural water is controlled by its Hg/DOC ratio. *Geochim. Cosmochim. Acta* **73**, 6704-
6715 (2009).

Fu, X. *et al.* Significant Seasonal Variations in Isotopic Composition of Atmospheric Total
Gaseous Mercury at Forest Sites in China Caused by Vegetation and Mercury Sources.
*Environ. Sci. Technol.* (2019).

Demers, J., Sherman, L., Blum, J., Marsik, F. & Dvonch, J. Coupling atmospheric mercury
isotope ratios and meteorology to identify sources of mercury impacting a coastal urban-
industrial region near Pensacola, Florida, USA. *Global Biogeochem. Cycles* **29**, 17 (2015).

Fu, X. *et al.* Isotopic Composition of Gaseous Elemental Mercury in the Marine Boundary
Layer of East China Sea. *J. Geophys. Res. - Atmos.* (2018).

Motta, L. C., Kritee, K., Blum, J. D., Tsz-Ki Tsui, M. & Reinfelder, J. R. Mercury Isotope
Fractionation during the Photochemical Reduction of Hg(II) Coordinated with Organic
Ligands. *J. Phys. Chem. A* **124**, 2842-2853, doi:10.1021/acs.jpca.9b06308 (2020).

Liu, C. *et al.* Sources and Transformation Mechanisms of Atmospheric Particulate Bound
Mercury    Revealed    by    Mercury    Stable    Isotopes.    *Environ    Sci    Technol,*





doi:10.1021/acs.est.1c08065 (2022).

Horowitz, H. M. *et al.* A new mechanism for atmospheric mercury redox chemistry:
Implications for the global mercury budget. *Atmos. Chem. Phys.* **17**, 6353-6371,
doi:10.5194/acp-2016-1165 (2017).

Motta, L. *et al.* Mercury Cycling in the North Pacific Subtropical Gyre as Revealed by
Mercury Stable Isotope Ratios. *Global Biogeochem. Cycles* **33**, 777-794 (2019).

Štrok, M., Baya, P. & Hintelmann, H. The mercury isotope composition of Arctic coastal
seawater. *C.R. Geosci.* **347**, 368-376 (2015).

Jiskra, M. *et al.* Mercury stable isotopes constrain atmospheric sources to the Ocean.
*Nature* **597**, 678-682 (2021).

Wang, Z. *et al.* Mass-dependent and mass-independent fractionation of mercury isotopes
in precipitation from Guiyang, SW China. *C.R. Geosci.* **347**, 358-367 (2015).

Sun, R. *et al.* Historical (1850–2010) mercury stable isotope inventory from anthropogenic
sources to the atmosphere. *Elementa-Sci. Anthrop.* **4**, 000091 (2016).

Gratz, L., Keeler, G., Blum, J. & Sherman, L. Isotopic Composition and Fractionation of
Mercury in Great Lakes Precipitation and Ambient Air. *Environ. Sci. Technol.* **44**, 7770
(2010).

Cantrell, C. A. Technical Note: Review of methods for linear least-squares fitting of data
and application to atmospheric chemistry problems. *Atmos. Chem. Phys.* **8**, 5477-5487
(2008).

60   A. Yamakawa, A. T., Y. Takeda, S. Kato and Y. Kajii. Emerging investigator series:
investigation of mercury emission sources using Hg isotopic compositions of atmospheric
mercury at the Cape Hedo Atmosphere and Aerosol Monitoring Station (CHAAMS), Japan.
*Environmental Science Processes & Impacts* (2019).

Akane Yamakawa, K. M., Jun Yoshinaga. Determination of isotopic composition of
atmospheric mercury in urban-industrial and coastal regions of Chiba, Japan, using cold
vapor multicollector inductively coupled plasma mass spectrometry. *Chem. Geol.* **448**, 9
(2017).

Kurz, A. Y., Blum, J. D., Johnson, M. W., Nadelhoffer, K. & Zak, D. R. Isotopic composition
of mercury deposited via snow into mid-latitude ecosystems. *Sci. Total Environ.* **784**,
147252, doi:10.1016/j.scitotenv.2021.147252 (2021).

Yu, B. *et al.* Katabatic Wind and Sea-Ice Dynamics Drive Isotopic Variations of Total
Gaseous Mercury on the Antarctic Coast. *Environ Sci Technol* **55**, 6449-6458,
doi:10.1021/acs.est.0c07474 (2021).

Yin, R. *et al.* Mercury Inputs to Chinese Marginal Seas – Impact of Industrialization and
Development of China. *Journal of Geophysical Research: Oceans* **123**, 5599-5611 (2018).

Zerkle, A. L. *et al.* Anomalous fractionation of mercury isotopes in the Late Archean
atmosphere. *Nat Commun* **11**, 1709, doi:10.1038/s41467-020-15495-3 (2020).

Guo, J. *et al.* Source identification of atmospheric particle-bound mercury in the
Himalayan foothills through non-isotopic and isotope analyses. *Environ. Pollut.* **286**,
117317, doi:10.1016/j.envpol.2021.117317 (2021).








Table 1 The possible Hg sources and atmospheric transformations for PBM
samples and the estimated contributions

| ID | X | Y | Z | $f_{red}$* (%) | $f_{red}$** (%) |
|----|----|----|----|----|----|
| Type i. GEM oxidation (negative $\Delta^{199}$Hg and near-zero $\Delta^{200}$Hg) | | | | | |
| 16 | 0.13 | -0.02 | 0.89 | | |
| 25 | 0.14 | -0.01 | 0.87 | | |
| 26 | 0.14 | -0.09 | 0.95 | N.A. | N.A. |
| 37 | 0.15 | -0.09 | 0.94 | | |
| Type ii. Photoreduction & troposphere photochemical reactions (positive $\Delta^{199}$Hg and $\Delta^{200}$Hg) | | | | | |
| 5 | 0.50 | 0.17 | 0.33 | | |
| 6 | 0.46 | 0.18 | 0.36 | | |
| 7 | 0.34 | 0.33 | 0.33 | | |
| 14 | 0.58 | 0.12 | 0.30 | | |
| 19 | 0.65 | 0.10 | 0.25 | | |
| 20 | 0.57 | 0.19 | 0.24 | | |
| 23 | 0.77 | 0.21 | 0.02 | | |
| 28 | 0.26 | 0.19 | 0.55 | 54±29 | 70±11 |
| 32 | 0.61 | 0.27 | 0.13 | | |
| 34 | 0.31 | 0.31 | 0.38 | | |
| 35 | 0.37 | 0.25 | 0.38 | | |
| 36 | 0.52 | 0.17 | 0.31 | | |
| 39 | 0.42 | 0.10 | 0.47 | | |
| 42 | 0.36 | 0.24 | 0.40 | | |
| 43 | 0.50 | 0.16 | 0.34 | | |
| Type iii. Photoreduction (positive $\Delta^{199}$Hg and near-zero $\Delta^{200}$Hg) | | | | | |
| 4 | 0.64 | -0.03 | 0.39 | | |
| 8 | 0.54 | -0.02 | 0.48 | | |
| 9 | 0.45 | -0.08 | 0.63 | | |
| 10 | 0.48 | 0.08 | 0.44 | | |
| 13 | 0.54 | -0.06 | 0.53 | | |
| 15 | 0.34 | 0.10 | 0.55 | | |
| 21 | 0.38 | 0.09 | 0.53 | | |
| 22 | 0.55 | 0.04 | 0.41 | | |
| 24 | 0.74 | -0.10 | 0.35 | 85±14 | 100±14 |
| 27 | 0.50 | -0.12 | 0.61 | | |
| 30 | 0.45 | 0.09 | 0.46 | | |
| 31 | 0.59 | 0.04 | 0.37 | | |
| 41 | 0.57 | 0.03 | 0.40 | | |
| 44 | 0.62 | 0.02 | 0.37 | | |
| 45 | 0.74 | -0.03 | 0.29 | | |
| 50 | 0.53 | -0.01 | 0.48 | | |
| 51 | 0.58 | 0.00 | 0.42 | | |





| | | | | | |
|---|---|---|---|---|---|
| Type iv. GEM oxidation & photoreduction (slightly negative $\Delta^{199}$Hg and near-zero $\Delta^{200}$Hg) | | | | | |
| 2 | 0.31 | 0.04 | 0.65 | | |
| 11 | 0.28 | 0.09 | 0.63 | | |
| 33 | 0.31 | -0.01 | 0.71 | N.A. | N.A. |
| 38 | 0.25 | 0.08 | 0.67 | | |
| Type v. troposphere photochemical reactions (negative $\Delta^{199}$Hg and positive $\Delta^{200}$Hg) | | | | | |
| 1 | -0.12 | 0.34 | 0.78 | | |
| 3 | 0.11 | 0.19 | 0.69 | | |
| 12 | -0.07 | 0.35 | 0.72 | | |
| 17 | 0.12 | 0.24 | 0.64 | | |
| 18 | -0.10 | 0.13 | 0.97 | -58±41 | -37±128 |
| 29 | 0.00 | 0.27 | 0.73 | | |
| 40 | -0.06 | 0.23 | 0.83 | | |
| 46 | 0.15 | 0.15 | 0.70 | | |
| 52 | 0.07 | 0.21 | 0.72 | | |
| Type vi. Anthropogenic emissions (slightly negative $\Delta^{199}$Hg and high Hg concentrations) | | | | | |
| 47 | 0.29 | -0.03 | 0.73 | | |
| 48 | 0.26 | 0.04 | 0.70 | N.A. | N.A. |
| 49 | 0.23 | 0.06 | 0.70 | | |
| mean | 0.37 | 0.10 | 0.53 | | |
| sd | 0.23 | 0.12 | 0.22 | | |

*The $f_{red}$ was calculated from ratios of $\Delta^{199}$Hg/$\Delta^{200}$Hg.
** The $f_{red}$ was calculated from X/(X+Y)

**Figures**

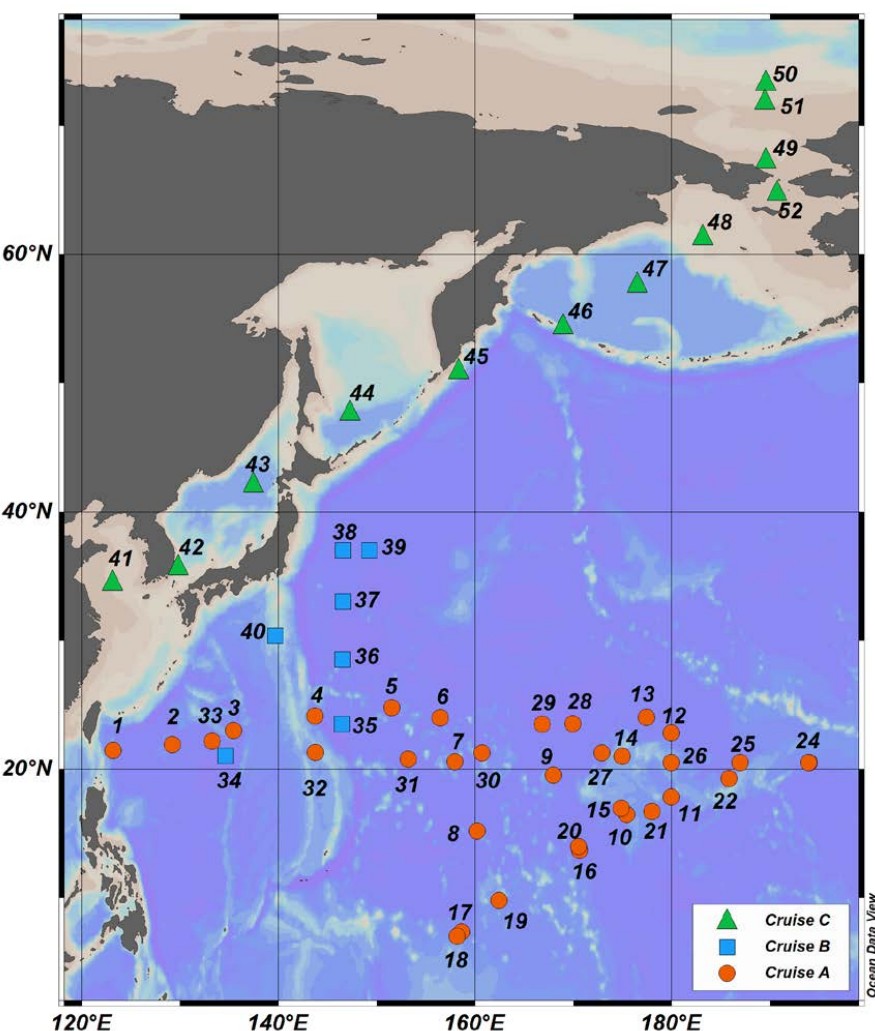

Figure 1. Map of sampling locations. The points represent the locations
at the "midpoint" moment during the sampling time of PBM samples.
(Ocean Data View 2020)



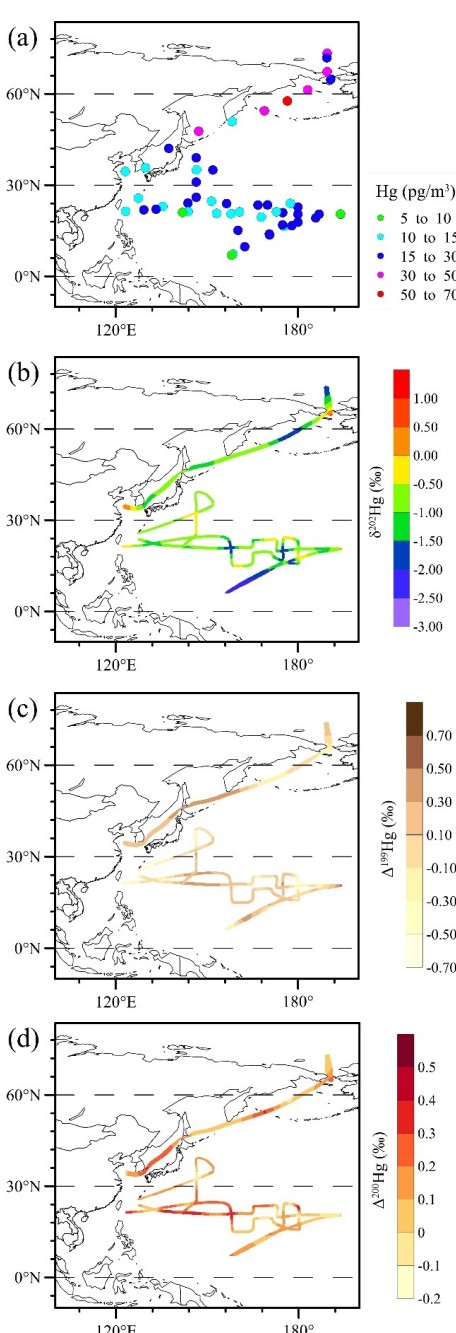

Figure 2. Spatial distributions of (a) Hg concentrations, (b) $\delta^{202}$Hg, (c) $\Delta^{199}$Hg, and (d) $\Delta^{200}$Hg of PBM in the MBL of the Northwest Pacific.



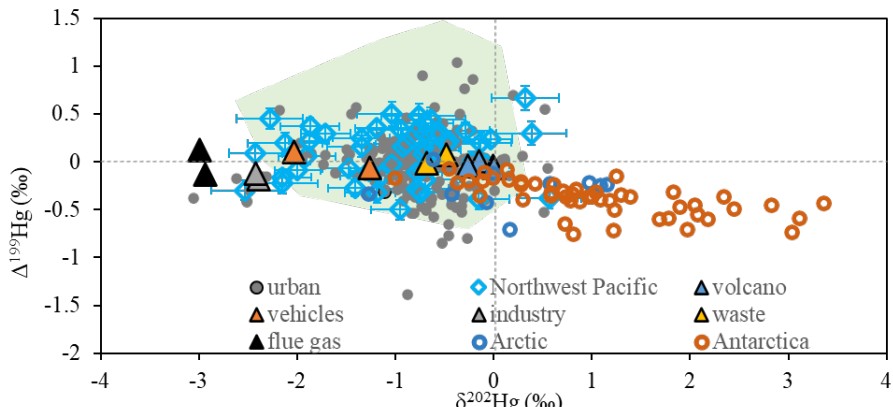

Figure 3. Plot of $\Delta^{199}$Hg vs. $\delta^{202}$Hg for PBM in the Northwest Pacific from this study, along with isotope data for PBM reported in the literature. The references for the literature data are: urban [19-23,37,38], volcano [40], traffic/vehicles [5], industry [5], waste [5], flue gas [39], arctic [13], antarctica [11], and high altitude areas (the green zone)[23,24].



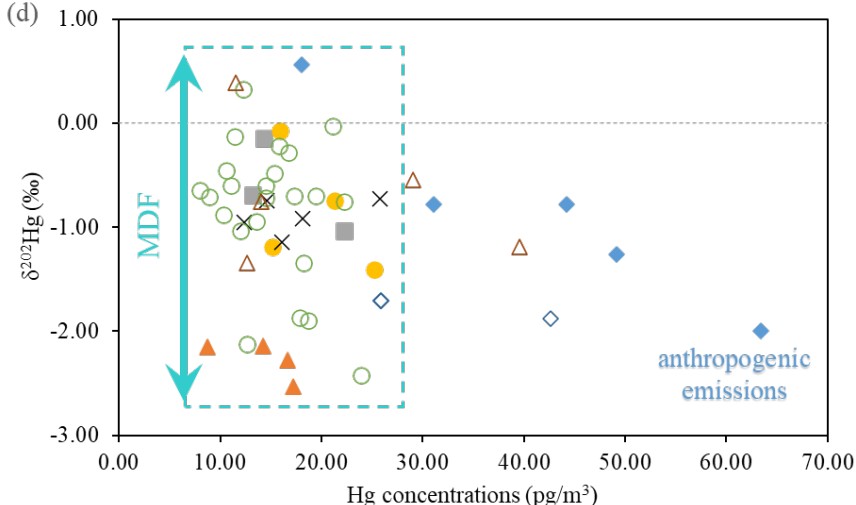

Figure 4. (a) Plot of $\Delta^{199}Hg$ vs. Hg concentration for PBM samples. (b) Plot of $\Delta^{199}Hg$ vs. Hg concentration for negative $\Delta^{199}Hg$ values of PBM during the different cruises. (c) Plot of $\Delta^{199}Hg$ vs. $\delta^{202}Hg$ for PBM samples. (d) Plot of $\delta^{202}Hg$ vs. Hg concentration for PBM samples. The error bar represents ±2SD for Hg concentrations. The typical 2SD analytic uncertainty of PBM samples was 0.22‰ to 0.42‰ and 0.07‰ to 0.13‰ for $\delta^{202}Hg$ and $\Delta^{199}Hg$, respectively, and listed in **Table S2**.





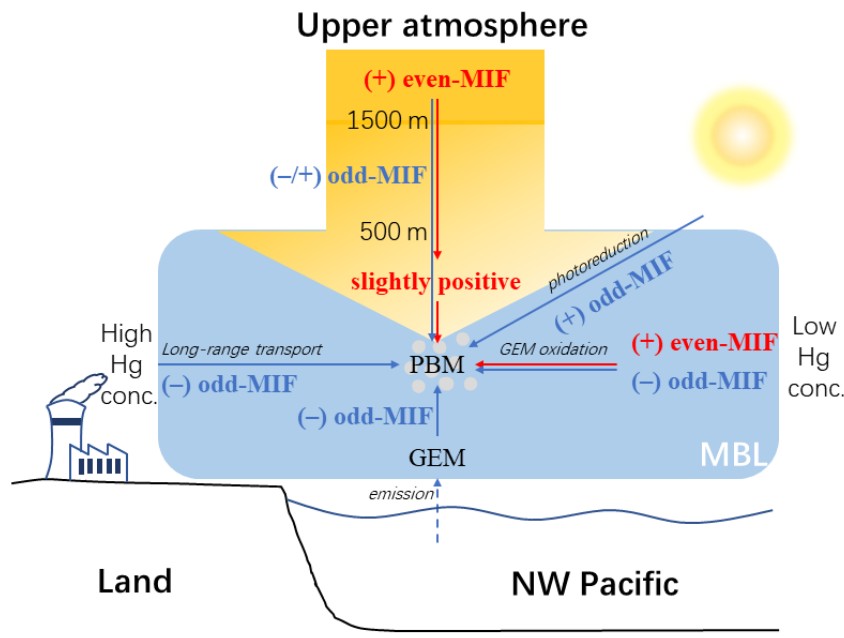

Figure 5. Schematic diagram of possible Hg sources and transformation processes for PBM over the MBL.



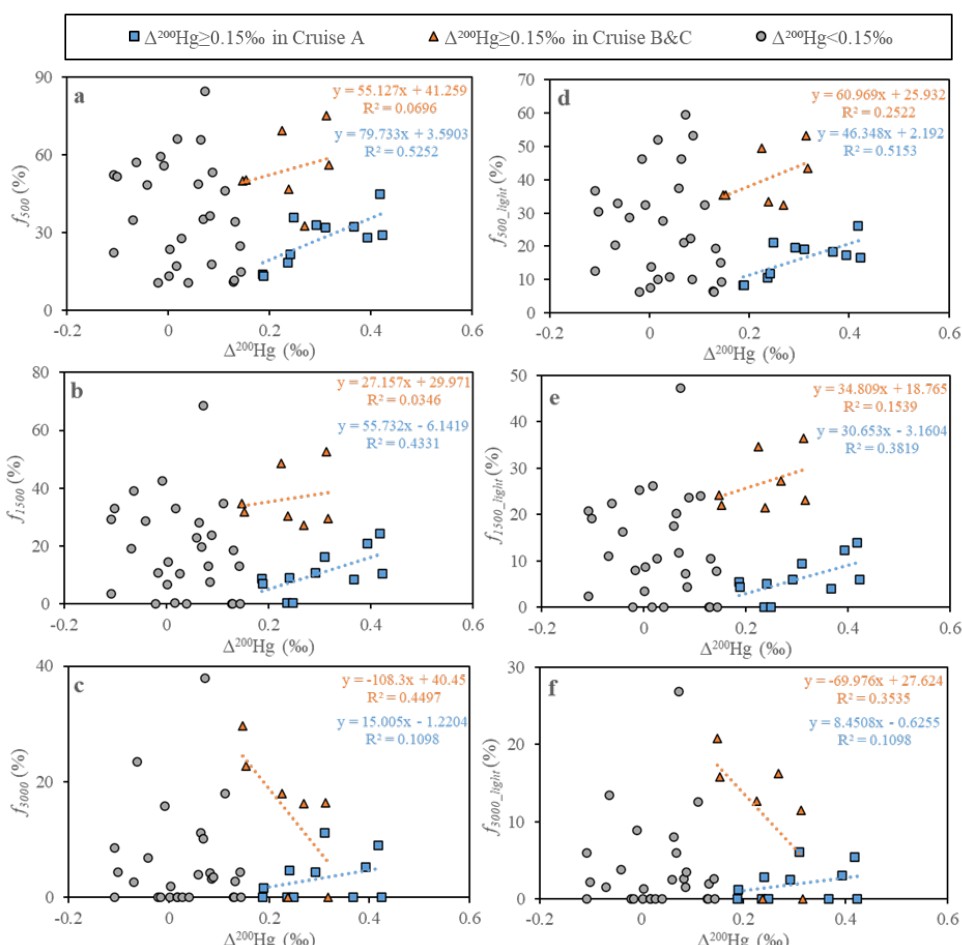

Figure 6. Plots of (a) $f_{500}$, (b) $f_{1500}$, (c) $f_{3000}$, (d) $f_{500\_light}$, (e) $f_{1500\_light}$, and (f) $f_{3000\_light}$ vs. $\Delta^{200}Hg$. The square and triangle represent PBM samples during Cruise A and Cruises B&C, respectively. The circles represent PBM samples with $\Delta^{200}Hg$ values lower than 0.15‰.

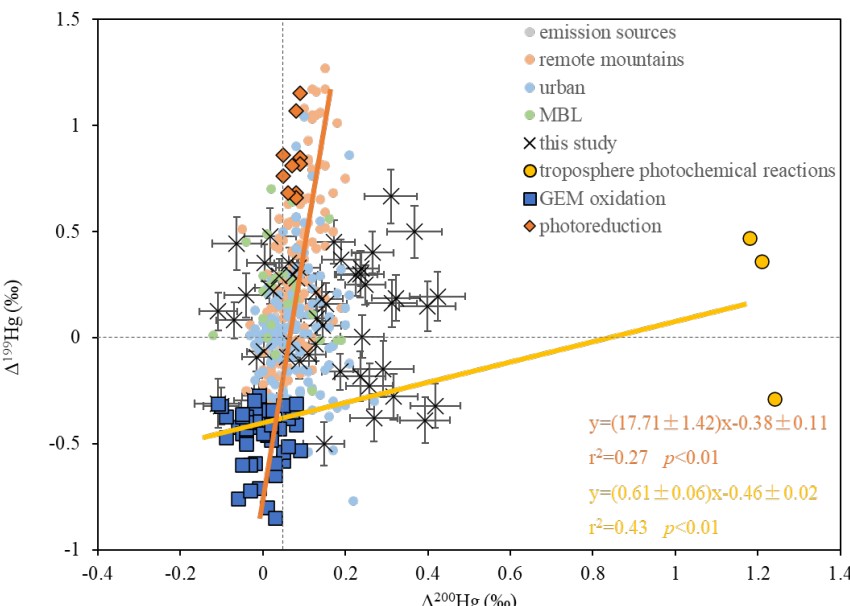

Figure 7. Plot of $\Delta^{199}$Hg vs. $\Delta^{200}$Hg for PBM samples collected from

emission sources [5,39,40], remote mountains [23,24,66], urban [19-23,38], MBL [17,27],

and this study. The three end-members of troposphere photochemical

reactions, GEM oxidation, and photoreduction are compiled from the

literature (**Table S5**). The criteria for the selection of the published Hg

isotope data are following: samples (mainly in remoted mountains) with

highly positive $\Delta^{199}$Hg (>0.66‰) and near-zero $\Delta^{200}$Hg values are

supposed to be dominated by photoreduction [24]; samples (mainly in

precipitation) with highly positive $\Delta^{200}$Hg (>1.10‰) values are supposed

to be dominated by troposphere photochemical reactions [8]; samples

(mainly in polar regions) with negative $\Delta^{199}$Hg (<-0.30‰) and near-zero

$\Delta^{200}$Hg values are supposed to be dominated by GEM oxidation [11,13,20,21,27].





Error bars represent the 2SD uncertainties of individual samples.