# Peer review of "Evidence for mass independent fractionation of even"

_Atmospheric Chemistry and Physics, 2022_

## Referee Comment (RC1)

Review of MS# acp-2022-284

Huang et al., *"Evidence for mass independent fractionation of even mercury isotopes in the troposphere"*

**General Comments:**

This study is aimed at understanding the origin of even MIF in Hg and suggests possible mechanisms. I find that the study doesn't harness the full potential of the dataset gathered. Furthermore, very poor writing combined with flawed approaches lead to weakened findings. Several sections are speculative, grammatically incorrect and lack appropriate referencing. The key aspects such as back trajectory analysis and statistical analysis lack methodological description and are not appropriately employed.

I do not recommend the publication of this manuscript in ACP.

**Specific Comments:**

Line 102 -109 : This section is poorly written and needs significant improvement. The HYSPLIT model-based air mass back trajectories shown in the supplement has not been referenced in the text. I suggest overlaying the back trajectories (Figure S6) with the cruise tracks in Figure 1.

While trajectory frequencies are useful, they are not direct enough to understand the air mass origins. I suggest the authors to conduct air mass cluster analysis using the trajectories. This would enable them to better segregate air mass origin source regions.

While computing the air mass trajectories ancillary information needs to be computed as well e.g., mixing layer height and trajectory height, rainfall etc. This will enable the authors to compute what fraction of the air mass transport occurred within the MBL. A table could be shown in the supplement.

Please note that the starting height of the trajectories (500 m) in Table S3 is too high. It should ideally be 50 m. Please ensure through cross examination if this height difference is affecting the back trajectories or not.

Line 111 -113 : Please provide more description of the sampling protocol. Why were two samplers used? Which samples were collected from which sampler at which heights ? Any blanks collected ? How many samples were collected in total ? How were the samples stored ? Which were the periods when sampling was stopped

I note that some of the information provided in the supplement has not been referenced in the text making it difficult for the reader. Every Note, Table and Figure in the Supplement should be referenced in the text in the main manuscript for enabling ease of reading.

Line 167 -168 : Why was the recovery > 100 % ?

Line 192- 202 : Very poorly written section. This needs significant improvement. I understand what the authors intend to do but it doesn't translate with enough fluidity on paper. Perhaps this point is referring to the point I made above regarding computing the fraction of air mass trajectory that is within the MBL. However, it is first important to know the height of the mixing layer. The cut-offs at 500m , 1500m, 3000m may be valid but not useful until the height of the mixing layer is known.

As this is a key component of the paper, I request the authors to please recompute the back trajectories (BT) with a starting height of 50 m with ancillary information for each BT. Then conduct cluster analysis to identify source regions. Following which, they should compute the fraction of BTs

within the mixing layer. These BTs should be shown in the main manuscript as an overlayed figure in Figure 1.

Line 204 -215: Again, a very poorly written section. This in my opinion is a key section of the methods. It is unclear what was the metric used to choose these three heights. The 'real' metric that should be used is the height of the mixing layer which the authors have not reported.

Furthermore, the Monte Carlo approach needs more description.

This choice will greatly affect the results and as such weakens the findings in the present form.

Line 269 -270 : This statement is not true. I would not refer to these poor correlations as strong in any manner. Why did the authors combine data for Cruise B&C in Figure 4b ? No justification has been provided for the same and given that they are entirely different cruise tracks and from different years which are clearly also affected by air masses from different origins, combining this dataset for a correlation plot is not correct.

Line 276 – 284: Too much speculation with no real proof. The authors do not provide a strong justification for their deduction.

Line 304 -320 : The justification for GEM oxidation pathway is not substantiated with any evidence. It is merely a speculation. Also, this section is poorly written.

Line 335: What is the abbreviation TGM ? It is not defined.

Line 313-314 and Figure 6: I find a fundamental flaw in the logic. The authors have self-imposed a cut off for the even-MIF and thereby left out a major chick of their data without any correlation in Figure 6. Also, why have the authors combined Cruise B &C data ? This approach to define even-MIF and further determine the mechanism is not convincing.

Along with this, the air mass frequencies need a real metric to determine the MBL transport and without the mixing layer height these self-imposed cut-offs (Line 382) yield speculative results.

Line 413- 417 : These numbers are speculative and self-imposed. A solid evidence is lacking and thereby weakens the discussion and main findings.

Line 490 -510 : The authors do not discuss at all how the end members were compiled and used. The authors should present the results from the Monte Carlo scheme used such as probability distribution functions and related plots. Overall in the present form the numbers are not convincing enough.

---

## Referee Comment (RC2)

General comments:

The paper's contribution is very commendable as it reports the isotope composition of particulate Hg in marine background air, where previously only the isotope composition in precipitation samples was reported. However, there are shortcomings, generalizations and less substantiated assumptions that require revision of the manuscript. The manuscript is surprisingly short and some parts need "more meat on the bones". Furthermore, the level of English should be improved and, above all, a reader of Atmos. Chem. Phys. must require that the authors lead a deeper discussion.

Many figures are busy and difficult to understand.

Specific comments:

The abstract is full with speculation.

L21 "The correlation between Δ200Hg and light conditions confirms that even-MIF is linked to photochemical reactions". If these refer to Fig. 6, several of these are weak and not always unambiguous. Please add confidence intervals. Why has the data from cruise B and C been combined? It is also difficult to explain that the highest Δ200Hg values tend to occur during cruise A when the light conditions were relatively low

L45 "dark abiotic redox reaction" The wording may seem inappropriate to express atmospheric transformation.

L46 Reference 2 is out of date and does not specifically apply to atmospheric transformation. The authors should cite publications that state the current paradigm regarding atmospheric redox processes, including Shat et al. EST 2021 etc.

L114 "To reduce the potential for contamination from the ship's exhaust plume, sampling was stopped during station work and when bad weather was encountered". This is a surprisingly crude procedure to segregate internal contamination. There must be windy conditions when the plume contaminated your samples. This can be seen, among other things, in CO/O3 ratios (if these were measured). Please comment on this.

L117 Isn't there a risk that GOM is caught on the filters and contributes to measured PBM values?

L134 it is significant disadvantage for a study that claims the troposphere as the source of even-MIF without measuring 204Hg. This information would have been invaluable.

L314 What is meant with RGM? Not defined.

L335 What is meant with TGM? Not defined.

L481 Awkward and vague. "Hg was photoreduced in oxidized Hg phases at high altitudes" Oxidized Hg phase???

L512 "The isotopic compositions of PBM in the MBL of the Northwest Pacific suggest that the even-MIF and odd-MIF signatures are useful tracers for identifying atmospheric transformations". The opening of the conclusion promises too much. What follows is pure speculation.

Fig 7. The selection of data for Fig. 7 is highly subjective. The referenced publications contain more data that is omitted. The interpretation of measurement data as examples of the proposed transformation paths is self-imposed and mis-leading.

---

## Author Comment (AC1)

General Comments:

This study is aimed at understanding the origin of even MIF in Hg and suggests possible mechanisms. I find that the study doesn't harness the full potential of the dataset gathered. Furthermore, very poor writing combined with flawed approaches lead to weakened findings. Several sections are speculative, grammatically incorrect and lack appropriate referencing. The key aspects such as back trajectory analysis and statistical analysis lack methodological description and are not appropriately employed.

I do not recommend the publication of this manuscript in ACP.

**Response: Thank you for your useful comments. The air mass back trajectories have been recomputed based on the HYSPLIT model according to your suggestion. More evidences and references have been added to support our data and analysis. We have asked professional language institutions to improve the grammar and writing of the full text.**

Specific Comments:

Line 102 -109: This section is poorly written and needs significant improvement. The HYSPLIT model-based air mass back trajectories shown in the supplement has not been referenced in the text. I suggest overlaying the back trajectories (Figure S6) with the cruise tracks in Figure 1.

While trajectory frequencies are useful, they are not direct enough to understand the air mass origins. I suggest the authors to conduct air mass cluster analysis using the trajectories. This would enable them to better segregate air mass origin source regions.

While computing the air mass trajectories ancillary information needs to be computed as well e.g., mixing layer height and trajectory height, rainfall etc. This will enable the authors to compute what fraction of the air mass transport occurred within the MBL. A table could be shown in the supplement.

Please note that the starting height of the trajectories (500 m) in Table S3 is too high. It should ideally be 50 m. Please ensure through cross examination if this height difference is affecting the back trajectories or not.

**Response: (Line 95) This section describes the three cruises in the Northwest Pacific. The irrelevant content (such as the air mass back trajectories) has been deleted.**
**Air mass cluster analysis has been conducted, and the trajectory frequencies in Fig. S2 have been replaced with the results of the air mass cluster analysis. Of course, we calculated the trajectories at the starting height of 50 m in this version of the manuscript.**
**The ancillary information, especially the mixing layer height, has been added and illustrated below.**

Line 111 -113: Please provide more description of the sampling protocol. Why were two samplers used? Which samples were collected from which sampler at which heights? Any blanks collected? How many samples were collected in total? How were the samples stored? Which were the periods when sampling was stopped

I note that some of the information provided in the supplement has not been referenced in the text making it difficult for the reader. Every Note, Table and Figure in the Supplement should be referenced in the text in the main manuscript for enabling ease of reading.

**Response: (Lines 112–116) Two samplers were deployed to ensure that sufficient samples were collected. The samples were collected at a height of approximately 15 m AMSL. The membranes collected from the two samplers were preconcentrated in the same trapping solution (see Section 2.3 and Supplementary Note 1). A description of the sampling blank has been added. A total of 105 membranes were collected and preconcentrated in 52 trapping solutions. The membranes and trapping solutions were preserved in cool (<4°C) and dark conditions. The periods during which sampling was stopped were not recorded as the sampler created an accurate record of the sampling duration. The information has been added to the main manuscript and Supplementary Table S1.**

Line 167 -168: Why was the recovery > 100% ?

**Response: (Lines 167–168) This could be due to either random error or the blank.**

Line 192- 202: Very poorly written section. This needs significant improvement. I understand what the authors intend to do but it doesn't translate with enough fluidity on paper. Perhaps this point is referring to the point I made above regarding computing the fraction of air mass trajectory that is within the MBL. However, it is first important to know the height of the mixing layer. The cut-offs at 500 m, 1500 m, 3000 m may be valid but not useful until the height of the mixing layer is known.
As this is a key component of the paper, I request the authors to please recompute the back trajectories (BT) with a starting height of 50 m with ancillary information for each BT. Then conduct cluster analysis to identify source regions. Following which, they should compute the fraction of BTs within the mixing layer. These BTs should be shown in the main manuscript as an overlayed figure in Figure 1.

**Response: (Lines 187-191) The back trajectories (BT) have been recomputed at a height of 50 m AMSL. The ancillary information for each BT has added to Table S4. Cluster analysis has been conducted, and the results are shown in Fig. S2. We thought overlaying the BTs with the cruise tracks in Fig. 1 would make it look chaotic, so we finally decided to present the BTs and cruise tracks separately in two figures.**

**(Lines 196–206) The average height of the mixed layer is 690 m and most of the sites had mixed layer heights of greater than 500 m although the height of the mixed layer varied with the sampling sites. In order to estimate the most active height of the even-MIF, we needed to fix the height of the mixed layer. In this case, we decided that 500 m was ideal and valid. In addition, 3000 m was set to represent the possible height at which the maximum $\Delta^{200}$Hg values were produced. This was determined based on the height of the rainfall because rainfall is characterized by the maximum $\Delta^{200}$Hg values reported so far.**

Line 204 -215: Again, a very poorly written section. This in my opinion is a key section of the methods. It is unclear what was the metric used to choose these three heights. The 'real' metric that should be used is the height of the mixing layer which the authors have not reported.

Furthermore, the Monte Carlo approach needs more description.

This choice will greatly affect the results and as such weakens the findings in the present form.

**Response: (Lines 419–426) As is discussed above, the height of the mixed layer would be more meaningful, but it varied with the sampling sites and could not be used for the estimation in this study. To explain the heights used, several sentences have been added. The isotope mixing model used to estimate the maximum $\Delta^{200}$Hg values is robust. Despite all of this, we had to remove the quantification results, and we need more even-MIF data to provide more accurate estimation. This research will be continuously improved in the future.**

Line 269 -270: This statement is not true. I would not refer to these poor correlations as strong in any manner. Why did the authors combine data for Cruise B&C in Figure 4b ? No justification has been provided for the same and given that they are entirely different cruise tracks and from different years which are clearly also affected by air masses from different origins, combining this dataset for a correlation plot is not correct.

**Response: (Lines 264–276) Thank you. The data for Cruises B and C have been separated. The poor correlations were mainly derived from the data for Cruise A, which was likely the result of the mixing of different Hg emission sources.**

*"The PBM samples with negative $\Delta^{199}$Hg values exhibit positive correlations with the Hg concentration (Fig. 4b). The samples collected during Cruises B and C exhibit stronger correlations ($r^2$=0.91, p<0.05 for Cruise B and $r^2$=0.87, p<0.05 for Cruise B) than the samples collected during Cruise A ($r^2$=0.04, p>0.1). The higher Hg concentrations of the PBM are most likely caused by long-range transport of anthropogenic emissions, which produced the slightly negative $\Delta^{199}$Hg values. The difference in the slopes of the correlations for Cruises B and C may be due to the different sources of anthropogenic Hg emissions, i.e., Japan and the Bering Sea, while the insignificant correlation for Cruise A is the result of mixing of various anthropogenic Hg emission sources (e.g., the South China Sea, Micronesia, and Hawaii)."*

Line 276 – 284: Too much speculation with no real proof. The authors do not provide a strong justification for their deduction.

**Response: (Lines 276–284) GEM oxidation with negative odd-MIF signatures has been observed in previous studies. This process could explain the samples with negative $\Delta^{199}$Hg values and lower Hg concentrations collected in this study. The lower Hg concentrations suggest that the PBM was not directly derived from anthropogenic emissions and that it experienced atmospheric transformation. Photoreduction generally enriches the reactant in odd isotopes, resulting in PBM with positive $\Delta^{199}$Hg values. Hence, the most likely cause of the negative odd-MIF is solely GEM oxidation.**

Line 304 -320: The justification for GEM oxidation pathway is not substantiated with any evidence. It is merely a speculation. Also, this section is poorly written.

**Response: Does the reviewer refer to the paragraph quoted below? There is nothing related to the GEM oxidation pathway. The main content is about the relationship between atmospheric transformation and MIF. Even-MIF is considered to be due to photochemical reactions in the upper troposphere and GEM oxidation at lower altitudes, while photoreduction is responsible for positive odd-MIF in PBM.**

*"In addition to odd-MIF, even-MIF is an important indicator of Hg sources and Hg transformation processes in the atmosphere [16]. As is shown in Fig. 5, the $\Delta^{200}Hg$ values were considered to be affected by photochemical reactions in the upper troposphere. GEM oxidation can also induce even-MIF at low altitudes, but it is limited to specific oxidants (e.g., Br and Cl). This process could be identified via backward air mass trajectory analysis, which is discussed below. Regarding the GEM oxidation processes without even-MIF, the resulting PBM generally inherits the even-MIF signatures of the GEM and RGM and is characterized by nearly zero $\Delta^{200}Hg$ values. Here, we conclude that the PBM dominated by troposphere photochemical reactions are characterized by odd-MIF and significant even-MIF ($\Delta^{200}Hg \geq 0.15‰$); and the PBM dominated by GEM oxidation without even-MIF at low altitudes are characterized by negative odd-MIF and nearly zero even-MIF at PBM. Therefore, except for the samples with high Hg concentrations (anthropogenic emissions), most of the PBM samples are related to photoreduction (+odd-MIF), GEM oxidation (–odd-MIF), and troposphere photochemical reactions (+even-MIF), which can be identified using the $\Delta^{199}Hg$ and $\Delta^{200}Hg$ values."*

Line 335: What is the abbreviation TGM ? It is not defined.

**Response: (Line 305) TGM denotes total gaseous mercury. It is now defined in the text.**

Line 313-314 and Figure 6: I find a fundamental flaw in the logic. The authors have self-imposed a cut off for the even-MIF and thereby left out a major chick of their data without any correlation in Figure 6. Also, why have the authors combined Cruise B &C data ? This approach to define even-MIF and further determine the mechanism is not convincing.
Along with this, the air mass frequencies need a real metric to determine the MBL transport and without the mixing layer height these self-imposed cut-offs (Line 382) yield speculative results.

**Response: The mixed layer height and the cut-offs at 500 m, 1500 m, and 3000 m have been explained above, and we hope that the explanation dispels the doubts of reviewer.**
**(Line 317) The cut off for the even-MIF ($\Delta^{200}Hg \geq 0.15‰$) is not self-imposed. It is derived from a previous study (Fu et al., 2019, ES&T). Additionally, in a previous review (Blum et al., 2014), the authors concluded that the $\Delta^{200}Hg$ anomaly should be larger than the typical analytical uncertainty (i.e., $>0.2‰$).**
**(Fig. 6) The reason we combined the data for Cruises B and C here is that the route of Cruise A was located south of 30°N, while the routes of Cruises B and C were primarily located north of 30°N. We used Cruise A to represent low latitudes and Cruises B and C to represent mid-high latitudes. For even-MIF, the latitudinal influence would be more important than the different origins of the air masses or the different years in which the cruises were conducted. In fact, the even-MIF of precipitation samples worldwide exhibit a**

**general increase with latitude, indicating that the factors causing even-MIF are likely related to latitude.**

Line 413- 417: These numbers are speculative and self-imposed. A solid evidence is lacking and thereby weakens the discussion and main findings.

**Response: (Lines 419–426) We have added references to support these numbers.**

Line 490 -510: The authors do not discuss at all how the end members were compiled and used. The authors should present the results from the Monte Carlo scheme used such as probability distribution functions and related plots. Overall in the present form the numbers are not convincing enough.

**Response: (Lines 503-524) The compiled criteria for the three end members are presented in the caption of Fig. 7. The Monte Carlo approach was not used in this method.**